# Upregulation of the Mevalonate Pathway through EWSR1-FLI1/EGR2 Regulatory Axis Confers Ewing Cells Exquisite Sensitivity to Statins

**DOI:** 10.3390/cancers14092327

**Published:** 2022-05-08

**Authors:** Charlie Buchou, Karine Laud-Duval, Wietske van der Ent, Sandrine Grossetête, Sakina Zaidi, Géraldine Gentric, Maxime Corbé, Kévin Müller, Elaine Del Nery, Didier Surdez, Olivier Delattre

**Affiliations:** 1INSERM U830, Équipe Labellisée LNCC, Diversity & Plasticity of Childhood Tumors Laboratory, PSL Research University, SIREDO Oncology Center, Institut Curie Research Center, 26 rue d’ULM, 75005 Paris, France; charlie.buchou@curie.fr (C.B.); karine.laud@curie.fr (K.L.-D.); w.van.der.ent@ncmm.uio.no (W.v.d.E.); sandrine.grossetete@curie.fr (S.G.); sakina.zaidi@curie.fr (S.Z.); didier.surdez@uzh.ch (D.S.); 2INSERM U830, Équipe Labellisée LNCC, Stress and Cancer Laboratory, PSL Research University, Institut Curie Research Center, 26 rue d’ULM, 75005 Paris, France; geraldine.gentric@curie.fr; 3Department of Translational Research, The Biophenics High-Content Screening Laboratory, PSL Research University, Institut Curie Research Center, 26 rue d’ULM, 75005 Paris, France; maxime.corbe@curie.fr (M.C.); kevinmuller@gmail.fr (K.M.); elaine.del.nery@curie.fr (E.D.N.); 4Balgrist University Hospital, University of Zurich, Zurich, Forchstrasse 340, 8008 Zürich, Switzerland

**Keywords:** Ewing sarcoma, MVA pathway, statin, new therapeutic strategy

## Abstract

**Simple Summary:**

The objective of this project was to search for new dependencies in Ewing sarcoma, a deadly disease for which new therapeutic approaches are urgently needed. A pharmacological screening of off-patent approved drugs (FDA agency) and the investigation of downstream targets of EGR2 were performed. The two approaches showed the MVA pathway as a major dependency in Ewing sarcoma and statin, an inhibitor of this pathway, as a potential new therapeutic agent for the treatment of Ewing sarcoma.

**Abstract:**

Ewing sarcoma (EwS) is an aggressive primary bone cancer in children and young adults characterized by oncogenic fusions between genes encoding FET-RNA-binding proteins and ETS transcription factors, the most frequent fusion being EWSR1-FLI1. We show that EGR2, an Ewing-susceptibility gene and an essential direct target of EWSR1-FLI1, directly regulates the transcription of genes encoding key enzymes of the mevalonate (MVA) pathway. Consequently, Ewing sarcoma is one of the tumors that expresses the highest levels of mevalonate pathway genes. Moreover, genome-wide screens indicate that MVA pathway genes constitute major dependencies of Ewing cells. Accordingly, the statin inhibitors of HMG-CoA-reductase, a rate-limiting enzyme of the MVA pathway, demonstrate cytotoxicity in EwS. Statins induce increased ROS and lipid peroxidation levels, as well as decreased membrane localization of prenylated proteins, such as small GTP proteins. These metabolic effects lead to an alteration in the dynamics of S-phase progression and to apoptosis. Statin-induced effects can be rescued by downstream products of the MVA pathway. Finally, we further show that statins impair tumor growth in different Ewing PDX models. Altogether, the data show that statins, which are off-patent, well-tolerated, and inexpensive compounds, should be strongly considered in the therapeutic arsenal against this deadly childhood disease.

## 1. Introduction

Ewing sarcoma (EwS) is an aggressive, highly metastatic bone tumor in children. While the survival of localized disease is around 70%, the prognostic of metastatic patients is still poor, with an overall 5-year survival below 25% [1,2]. Many efforts are presently being undertaken to more thoroughly investigate the mechanisms of development of EwS in order to identify fragilities and dependencies for new targeted therapeutic interventions.

EwS is caused by a chromosomal translocation that fuses the N-terminal prion-like domain of the EWSR1 protein with the C-terminal DNA-binding domain of an ETS transcription factor, most frequently FLI1 [2,3]. The resulting chimeric protein behaves as an aberrant transcription factor that dysregulates the expression of a variety of target genes, particularly through its neomorphic ability to generate neo-enhancers at GGAA microsatellite repeats [4,5,6,7,8]. Other genetic alterations are rare in EwS and mostly involve STAG2, TP53, and CDKN2A at low frequencies [9,10].

The highly variable frequency of EwS from one human population to another has long suggested a germline genetic susceptibility to this disease. To further explore this susceptibility, we recently conducted genome-wide association studies (GWASs) which identified six highly significant susceptibility loci [11,12]. One of these loci encodes EGR2/KROX20, which was further shown to be a cis-mediated target of EWSR1-FLI1 and to be essential for EwS growth in vivo in xenograft models. One of the polymorphisms observed in EwS patients was shown to increase the length of a GGAA microsatellite repeat motif, thus inducing increased binding of EWSR1-FLI1 and, consequently, increased EGR2 expression [13]. This study therefore pointed out the key role of EGR2 expression in the development of EwS.

EGR2 is a transcriptional factor [14] that is essential for hindbrain development and for the myelination of the peripheral nervous system by Schwann cells [15,16]. This last function requires large amounts of lipid and cholesterol biosynthesis [17]. EGR2 was further shown to activate key genes involved in the myelinization program, including cholesterol/lipid biosynthetic genes [17].

Here, we developed two orthogonal strategies to search for new dependencies in EwS: one based on the investigation of downstream targets of EGR2, the other relying on a pharmacological screening of off-patent approved drugs (FDA, EMA, and other agencies). The first strategy identified the mevalonate pathway, including HMGCR and HMGCS1, which encode the two rate-limiting enzymes of this pathway, as a major target of EGR2, while the latter identified statins, which are well-known inhibitors of HMGCR, as major cytotoxic compounds. These two approaches hence converged to point out the MVA pathway as a critical dependency in EwS. We further show here that statin treatment led to a slowdown of the S-phase and increased apoptosis of EwS cells through increased levels of oxidative stress and the modulation of protein prenylation. Finally, using in vivo experiments with PDX models, we document the anti-tumoral effect of statin treatment.

## 2. Materials and Methods

### 2.1. Cell Culture, EWSR1-FLI1, and EGR2 Invalidation

A673, SKNMC, TC71, POE, CHLA10, MHHES1, and EW1 are patient-derived Ewing sarcoma cell lines. The Ewing sarcoma A673 and SKNMC cell lines were obtained from the American Type Culture Collection (ATCC), the Ewing sarcoma TC71 and MHHES1 cell lines from the German Collection of Microorganisms and Cell Cultures (DSMZ), and the EW1 cell line from the International Agency for Research on Cancer (IARC). The CHLA10 cell line was obtained from the Childhood Cancer Repository (COG Repository; MTA#20150093). The A673/TR/EF1 clone, deriving from the A673 cell line, was provided by J. Alonso (Madrid, Spain [18]). EWIma1 was generated by engineering an EWSR1-FLI1 translocation in MSCs [19]. The POE shEGR2_4#22 clone was generated with the POE cell line (previously described by [13]); it harbors a doxycycline-inducible shRNA against EGR2. The A673 cell line and A673/TR/EF1 clone were maintained in DMEM (SH30022.01; GE healthcare, Chicago, IL, USA) with 10% fetal bovine serum (Sigma-Aldrich, St. Louis, MO, USA) and 1% Penicillin/Streptomycin (Thermofisher Scientific, Waltham, MA, USA); Blasticidin (10 μg/mL; Merk, Darmstadt, Germany) and Zeocin (100 μg/mL; Invitrogen, Waltham, MA, USA) were added to A673/TR/EF1 at 37 °C with 5% CO_2_. SKNMC, TC71, MHHES1, EW1, and POE cells and the POE shEGR2_4#22 clone were cultured in RPMI (SH30027.01; GE Healthcare), 10% FBS, 1% Penicillin/Streptomycin, and 5% CO_2_, and on collagen from bovine skin (C4243; Sigma-Aldrich, St. Louis, MO, USA) for SKNMC, TC71 and EW1. CHLA10 cells are cultured in IMDM (12440-053; Gibco, Waltham, MA, USA) supplemented with 20% FBS, 4 mM L-Glutamine (25030-024; Gibco), and 1X ITS (5 µg/mL insulin, 5 µg/mL transferrin, 5 ng/mL selenous acid; 41400-045; Gibco). Inhibition of EWSR1-FLI1 in the A673/TR/EF1 clone and of EGR2 in the shEGR2_4#22 clone was induced by treating with 1 μg/mL doxycycline. Inhibition of EWSR1-FLI1 or EGR2 in A673, SKNMC, TC71, POE, CHLA10, MHHES1, and EWIma1 was achieved by transfection with 50 nM of the siRNA targeting EWSR1-FLI1, 5′-AAGGCAGCAGAACCCTTCTTA-3′; with 15 nM of #1 5′CACCTAGAAACCAGACCTTCA3′ and #2 5′GCTACCCAGAAGGCATAATCAATAT3′, targeting EGR2; or EWSR1-FLi1 type 2, 5′-GGCAGCAGAGTTCACTGCTCG-3′, for EW1 cells using RNAiMAX Reagent (Thermofisher Scientific) according to the manufacturer’s instructions. MCF-7 (breast carcinoma), SKNSH (neuroblastoma), A549 (lung carcinoma), and KD (Rhabdoid) were cultured in DMEM or RPMI for KD supplemented with 10% FBS (F7524; Sigma-Aldrich) at 37 °C and 5% CO_2_. Cells were checked routinely by PCR for the absence of mycoplasma (VenorGeM qEP; Minerva Biolabs, Berlin, Germany). Cell authentication was performed by comparison with STR profiling from ATCC (A673, SK-N-MC) and DSMZ (TC71).

### 2.2. Gene Expression Profile

The differential analysis of genes modulated by EWSR1-FLI1 and EGR2 was performed with data obtained with Affymetrix Human Gene 2.1 ST arrays as described by [13]. The data were deposited at the Gene Expression Omnibus (GEO; GSE62090). The differential expression of mevalonate pathway enzymes and prenyltransferases (in Green) in 6 Ewing cell lines was performed with RNAseq data obtained after invalidation of EWSR1-FLI1 by siEF1 at different time points. The data were deposited at the Gene Expression Omnibus (GSE 132966, GSE 156653, GSE 150777, and GSE 164373).

### 2.3. Chemical Compounds

For in vitro testing, Atorvastatin calcium, Simvastatin, and Lovastatin were solubilized in DMSO at 10 mM stock solution (s2077, s1792, and S2061; Selleckchem); for in vivo experiments, Atorvastatin calcium was prepared in 4% DMSO, 35% PEG300 (Sigma-Aldrich), and 2% Tween 80 (Sigma-Aldrich). Farnesyl-pyrophosphate (Fpp) (2 mM), Geranylgeranyl-PP (GGpp) (2 mM), Mevalonolactone or Mevalonic acid (MEV), and N-Acetyl Cysteine (NAC) were purchased from Sigma-Aldrich. MEV and NAC were solubilized, respectively, in ethanol 95% (100 mM) and in water (100 mM). The Prestwick Chemical Library (hereafter PCL) is a library of 1280 off-patent small molecules, mostly approved drugs (FDA, EMA, and other agencies). The compounds were provided at a concentration of 10 mM in 100% DMSO.

### 2.4. High-Throughput Drug Screening and Hit Calling

The high-throughput drug screening was performed using the PCL on the BioPhenics platform of Institut Curie, Paris, France (Elaine Del-Nery). The PCL V3 (1280 small molecules) and V2 (1200 small molecules) off-patent approved drugs were selected for their bioavailability and safety for humans. For the screening, the A673 cell line was seeded in 384-well plates the day prior to compound addition. The next day, a single dilution of each compound of PCL V2 and V3 at 10 µM was added to the cells for 24 h and 72 h, respectively. After fixation with 4% formaldehyde solution, cell nuclei were stained with 0.2 µg/mL of DAPI and image acquisition was performed using the IN Cell 2200 automated wide-field microscopy system (GE Healthcare) at 10× magnification. Nucleus counts were performed using a specific module implemented in IN Cell Analyzer Workstation software (GE Healthcare). Raw cell-count values were obtained by the average of 4 image fields per well and transformed in log function to obtain a robust Z-score-standardized robust estimator value, as previously described [20]. The compounds with an RZ-score < −2 correspond to decreased cell proliferation. Two biological replicates were obtained for the 24 h time-point experiment and three for the 72 h time-point experiment. Data analysis and hit-calling details are provided in Appendix A.

### 2.5. Western Blot Analysis

Total proteins were extracted using RIPA buffer (150 mM NaCl, 50 mM Tris pH 7.5, 1 mM EDTA, 0.1% SDS, 0.25% deoxycholic acid, 1% NP-40, 1 mM PMSF, and protease inhibitor cocktail tablets (Roche, Basel, Switzerland)). Whole cell lysates were resolved by SDS-PAGE and transferred onto a nitrocellulose membrane (Bio-Rad, Hercules, CA, USA). Blots were incubated with the following antibodies: rabbit anti-FLI1 (1:1000; ab133485; Abcam, Cambridge, UK), rabbit anti-EGR2 (1:1000; ab108399; Abcam), rabbit anti-HMGCS1 (1:2000; ab194971; Abcam), rabbit anti-HMGCR (1:3000; ab174830; Abcam), rabbit anti-MVD (1:3000; ab129061; Abcam), rabbit anti-MVK (1:3000; ab126619; Abcam), rabbit anti-caspase3 (1:1000; #9662; Cell Signaling, Danvers, CA, USA), rabbit anti-cleaved caspase-3 (1:1000; #9661; Cell Signaling), rabbit anti-PARP (1:1000; #9542; Cell Signaling), rabbit anti p21 (1:500; #2947; Cell signaling), and rabbit anti-HRP-conjugated GAPDH (1:10,000; HRP-60004; ProteinTech, Manchester, UK); blots were probed with the corresponding immunoglobulin horseradish peroxidase (HRP) conjugated secondary antibodies (1:3000; NA934; Sigma-Aldrich). Proteins were visualized using SuperSignal West Pico Plus (34580; Thermofisher Scientific) and ChemidocTM Imaging System (Bio-Rad). Signal intensity quantification was measured with Image lab software (Bio-Rad). Original Western Blot figures shown in Appendix A.

### 2.6. ChIP-Seq

Chromatin Immunoprecipitation (ChIP) experiments were performed with the POE shEGR2_4#22 clone stably expressing the EGR2 protein fused to the HA tag. Previously, endogenous EGR2 protein expression was inhibited by doxycycline-inducible shRNA against EGR2 for 5 days. ChIP was conducted following the manufacturer’s protocol using the iDeal ChIP-seq kit for transcription factors and histones (Diagenode; Denville, NJ, USA) with anti-HA (2 µg; Y11; Santa-Cruz), rabbit polyclonal anti-Fli (2 µg; 15289; Abcam), rabbit polyclonal anti-H3K4me3 (1 µg; C15410003; Diagenode), and rabbit polyclonal anti-H3K27ac (1 µg; ab4729; Abcam). ChIP sequencing was performed on Illumina HiSeq 2500 using 100 bp single-end sequencing; libraries were generated using the TruSeq ChIP library preparation kit (Illumina, San Diego, CA, USA). The obtained reads were aligned in the human reference genome (GRCh37/hg19) using bowtie2-2.2.9 [21]. Uninformative reads (multimapped reads, duplicated reads, and reads with low mapping score) were filtered out with samtools 1.3 [22]. Peaks were called with MACS2 2.1.1 [23] with the narrow option for EGR2 ChIP-seq and the broad option for histone marks. ChIP-seq were normalized according to the input DNA sample. The ChIP-seq signal tracks were generated by macs2 with the bdgcmp option (and –m FE to compute fold enrichment between the ChIP and the control). Then, we ran bedGraphToBigWig to convert the file to a binary format (BigWig).

### 2.7. Ewing PDX Tumor-Dissociated Cells

Four Ewing sarcoma PDX models, IC-pPDX-80, IC-pPDX-87, IC-pPDX-3 [24], and IC-pPDX-164, were generated at Institut Curie from patients under an Institutional-Review-Board-approved protocol (OBS170323CPP ref3272; dossier No. 2015-A00464-45). All PDX tumors exhibited a type 1 fusion transcript of EWSR1-FLI1 (EWSR1 ex7/FLI1 ex6 fusion) except for IC-pPDX-164, showing EWSR1-FEV fusion. The dissociation of PDX tumors into single-cell suspensions was performed as previously described in [24], except that the enzymatic dissociation was realized in CO_2_-independent medium (GIBCO) containing 150 µg/mL Liberase (Roche) and 150 µg/mL DNase (Roche), for 30 min at 37 °C with gentle mixing. The single cells were resuspended in DMEM-F12 (Thermofisher Scientific) supplemented with B-27 serum-free supplement 1× (#17504044; Thermofisher Scientific). Cellular viability was quantified using a Vi-cell XR Viability Analyzer (Beckman Coulter, Brea, CA, USA).

### 2.8. Cell Proliferation in 2D and 3D Cultures (Spheroid Assay) and IC50 Determination

The assessment of the effect of Atorvastatin and Simvastatin on cell proliferation and the IC50 determination were conducted by seeding, respectively, 5 × 10^4^ A673, TC71, MCF7, A549, SKNSH, and KD, 7.5 × 10^4^ POE, and 10 × 10^4^ SKNMC in each well of 24-well plates. For PDX-dissociated cells, 14 × 10^4^ cells were seeded per well. After 24 h, a serial dilution of statins (30 µM->0.12 µM) was added for 72 h for the treatment of the cell lines and for 6 days for PDX-dissociated cells. Cell counting was performed by the trypan blue exclusion method using a Vi-cell XR Viability Analyzer (Beckman Coulter). IC50 values were determined by nonlinear fit dose–response curves using prism graphpad 8 software, (San Diego, CA, USA) and normalized to cells treated with DMSO.

For the 3D culture, single spheroids per well were formed from 800 dissociated cells derived from a PDX tumor in a low ultra-attachment 96-well plate (ULA plate; 7007; Corning). After 4 days of spheroid formation, the serial dilution of statins (30 µM->0.12 µM) was added for 7 days. Images of spheroids were taken on a Leica microscope with a DFC 3000G camera at 10× magnification. The growth of spheroids was measured on day 7 by area calculation using imageJ software. All experiments were performed in duplicate. The measure of the surface area of spheroids was normalized to spheroids treated with DMSO.

### 2.9. Cell-Cycle Analysis and Annexin V Staining

A673 (3.5 × 10^5^ cells), TC71 (4 × 10^5^ cells), and POE (6 × 10^5^ cells) were seeded in flasks of 25 cm^3^. After 24 h, the cells were treated with, respectively, 1 µM, 2.6 µM, and 2.4 µM Atorvastatin for 72 h. Cell-cycle analysis was performed by EdU staining (5′ethylnyl-2′-deoxyuridine) with 10 µM EdU solution for 30 min at room temperature. The cells were harvested, washed with PBS, resuspended in 0.5% BSA-PBS, and fixed with 70% ethanol overnight at 4 °C. Then, the cells were centrifuged and permeabilized with 0.2% Triton X-100 for 20 min. The detection of EdU labeling was performed with azide coupled with 1.8 µL of Alexa Fluor 647 dye (C10419; ThermoFisher Scientific) in PBS buffer containing 2 µM CuSO4 and 10 µM Sodium Ascorbate for 30 min at room temperature in the dark. The cells were stained with PI solution (0.5 µg/mL Pi and 50 µg/mL RNAseA) and analyzed on an LSRII flow cytometer (Becton Dickinson, Franklin Lakes, NJ, USA). Cell-cycle progression was studied using an EdU pulse–chase experiment on asynchronized cells, and the Double Thymidine block (DT) method was used to synchronize cells at the G1/S boundary, with the A673 and TC71 cell lines having been treated for 72 h with Atorvastatin at, respectively, 1 µM and 4.6 µM, corresponding to 80% of cell proliferation inhibition. For the EdU pulse–chase, cells were pulsed with EdU (10 µM) for 30 min, washed, and released in fresh media. EdU-positive cells and unstained cells were followed overtime (2 h, 4 h, 6 h, 8 h, and 10 h) through the phases of the cell cycle by flow cytometry. For the DT analysis, 2 mM thymidine was added to the culture medium for 14 h; the cells were washed twice with culture medium and covered with medium for 9 h. Then, 2 mM thymidine was added to the culture for 14 h. At the end of the second round of thymidine treatment, the cells were washed and collected at 0, 2 h, and 4 h after release. Cells were stained with EdU (10 µM) for 30 min each time they were collected.

For the apoptosis analysis, an Annexin-V-FITC/PI Apoptosis Detection Kit I (Becton Dickinson) was used according to the manufacturer’s protocol. All samples were assayed on an LSRII flow cytometer (Becton Dickinson) and analyzed with FlowJo software. Annexin-V-positive cells were normalized to the percentage of Annexin-V-positive cells treated with DMSO. The cells were treated with 0.5 µM (A673), 2.6 µM (TC71), and 2.4 µM (POE) Atorvastatin for 72 h.

### 2.10. Rescue Experiments on Cell Proliferation

The implication of the mevalonate pathway in the cytotoxic effects of statins on Ewing cell lines and PDX-dissociated cells was assessed by rescue experiments performed by the addition of intermediate products of lipid/cholesterol synthesis. Three Ewing cell lines (A673, TC71, and POE) were seeded in a 24-well plate as described previously. After 24 h, Atorvastatin (1 µM A673, 2.6 µM TC71, and 2.4 µM POE) was added to the culture medium in the absence or presence of 100 µM MEV, 10 µM Fpp, or GGpp for 72 h. The experiments with NAC, an inhibitor of intracellular ROS (NAC), were performed in flasks of 25 cm^3^ with 4.5 × 10^5^ of A673 cells. A total of 24 h after being seeded, cells were pre-treated for 2 h with NAC at 3 mM before the addition of Atorvastatin (1 µM) and collected 48 h post treatment. Cell viability was measured with the trypan blue exclusion method using a Vi-cell XR Viability Analyzer (Beckman Coulter). All analyses were conducted in duplicate in 3 independent experiments and normalized to cells treated with DMSO.

### 2.11. Intracellular Cell ROS and Lipid Peroxidation Measurement

The intracellular ROS level was studied using a Probes CellROX Deep Red kit (C10422; ThermoFisher Scientific) and lipid peroxidation using Bodipy 580/591 C11 (Lipid peroxidation sensor) (D3861; ThermoFisher Scientific). Briefly, the A673, TC71, and POE cell lines were seeded at 1 × 10^5^ cells per well, except for the POE cell line (2 × 10^5^), in 6-well plates, treated with Atorvastatin and stained with 5 µM Cell Rox Dep Red probes or 2 µM Bodipy C11 for 30 min at 37 °C, 5% CO_2_. Then, cells were washed with 1 mL of PBS 1×, harvested and resuspended in 200 µL of PBS1X complemented with 10% of FBS and 2.5 µg/mL DAPI solution. Cells were analyzed by flow cytometry using an LSR FORTESSA analyzer (BD Biosciences, Franklin Lakes, NJ, USA). The induction of intracellular ROS by Atorvastatin was assessed with 2 doses of Atorvastatin inducing 50% and 80% of proliferation inhibition after 72 h of treatment (A673, 0.5 µM and 1 µM; POE, 2.4 µM and 4.8 µM; TC71, 2.6 µM and 5.2 µM).

For the study of the mevalonate pathway’s role in the modulation of intracellular ROS levels and lipid peroxidation induced by statins, the A673 cell line was treated with 500 nM Atorvastatin and the rescue experiments were performed with MEV (100 µM), Fpp (10 µM), and GGpp (10 µM) for 72 h. For the measurement of ROS upon NAC treatment, the A673 cell line was seeded at 1.5 × 10^5^ and pretreated with 3 mM NAC, followed by the addition of Atorvastatin (1 µM).

All experiments were performed in duplicate, in 3 independent experiments. Mean fluorescence intensity (MFI) was performed with flowjo 9.8.2 software (BD Biosciences) and relative specific MFI was calculated by normalizing to cells treated with DMSO.

### 2.12. Prenylated Protein Extraction

Prenylated proteins were separated using the Triton X-114 phase-separation method according to [25]. Briefly, cells were resuspended in PBS 2% pre-condensed Triton X-114 for 30 min at 4 °C, followed by centrifugation at 13,000 rpm for 10 min at 4 °C. Phase separation was achieved by incubating at 37 °C for 10 min, followed by centrifugation at 13,000 rpm for 10 min at room temperature. The detergent phase containing prenylated protein was collected and washed 2 times using PBS 0.1% pre-condensed Triton X-114. The proteins were extracted with the Methanol/chloroform method (as described by [25]) by adding 1/5 volume of methanol and 4/5 volume of chloroform for 20 min to the detergent phase. After centrifugation at 13,000 rpm for 30 min at 4 °C, the upper phase was removed and 9 volumes of methanol was added to the lower phase, followed by centrifugation at 13,000 rpm for 30 min at 4 °C. The pellet was resuspended in Laemli Buffer containing 1 mM DTT. The detergent phase was resolved by SDS-PAGE and transferred onto a nitrocellulose membrane (Bio-Rad). Blots were incubated with the antibodies rabbit anti-RhoA (1:1000; #2117; cell signaling), mouse anti-Rac1 (1:500; 610650; BD Biosciences), rabbit anti-Ras (1:10,000; ab52939; abcam), and goat anti-HLA-A (1:500; sc-23446; Santa Cruz) and the secondary antibodies HRP conjugated mouse anti-goat IgG (1:10,000; sc-2354; Santa Cruz) and HRP conjugated sheep anti-mouse IgG (1:3000; NA931; Sigma Aldrich).

### 2.13. Xenograft of Ewing PDX Models

Tumor fragments of IC-pPDX-87, IC-pPDX-3, and IC-pPDX164 were grafted into the fat pad of 12 mice per PDX model. At a tumor size of around 32–60 mm^3^, mice were randomly assigned to Atorvastatin or DMSO groups. Mice were treated every day by intraperitoneal injection of DMSO or Atorvastatin (10 mg/kg/day) until one mouse reached the ethic size volume of 2000–2500 mm^3^, measured by the formula V = D × d^2^/2, where D is the largest diameter and d the smallest one. Tumor volume was measured every 2 or 3 days with a caliper.

This study was performed in accordance with the recommendations of the European Community (2010/63/UE) for the care and use of laboratory animals. Experimental procedures were approved by the ethics committee of Institut Curie CEEA-IC #118 (Authorization APAFIS#11206-2017090816044613-v2 given by National Authority) in compliance with the international guidelines.

### 2.14. Statistics

IC50 values were determined by nonlinear fit dose–response curves using prism graphpad 8 software and normalized to cells treated with DMSO. All statistic tests except for Bayesian Student’s *t*-tests were performed using prism graphpad 8 software. Bayesian Student’s *t*-tests were performed using the Limma R package. Boxplots are represented from min to max with median representation. Barplots, growth curves, and last-day tumor volume are represented as mean +/− SEM when the repliquat number between cell lines was different, otherwise +/− SD was used. EwS cells’ higher sensibility to statin was analyzed using Bayesian *t*-tests as used in depmap portal high-throughput screening. The statistical evaluation of Mevalonate pathway expression was performed with two-tailed unpaired Student’s *t*-tests with Welch’s correction. The Atorvastatin effect or rescue effect was analyzed using two-tailed unpaired Student’s *t*-tests, whereas a two-tailed paired Student’s t-statistic was used to access the in vivo growth inhibitory effect of Atorvastatin, and a comparison of the mean volume of the tumor on the last day treatment was performed using a nonparametric Wilcoxon rank test. The *p*-value was considered significant at *p* < 0.05.

## 3. Results

### 3.1. EWSR1-FLI1 and EGR2 Regulate Lipid/Cholesterol/Mevalonate Pathway

We previously showed that EWSR1-FLI1 directly regulates the expression of the EGR2 susceptibility gene in Ewing cells by binding to a GGAA repeat motif [12,13]. As EGR2 encodes a transcription factor that is essential to Ewing oncogenesis [13], we undertook a characterization of its mechanisms of action by investigating downstream targets. Upon downregulation of EGR2 with two different siRNAs (siEGR2 #1 and #2) in the A673 and SKNMC Ewing cell lines, we found a common set of 462 modulated genes (log2FC > 0.5) upon EGR2 knock-down (EGR2-KD) in both cell lines (Figure 1A). Genes down-regulated upon EGR2-KD (280 genes) demonstrated a very significant enrichment in genes involved in the cholesterol/steroid synthesis pathway (Figure 1B and Appendix A). This result is in full agreement with the previously identified central function of EGR2 in myelinization programs [15,16] and with its direct role in the regulation of the lipid/cholesterol biosynthetic pathway [17]. Of particular interest was the presence of a set of seven genes highlighting the role of EGR2 in the regulation of the mevalonate (MVA) pathway. This pathway is key for the biosynthesis of cholesterol and sterol biosynthesis but also for the production of isoprenoid lipids (Fpp, Farnesyl pyrophosphate, and GGpp, Geranyl-Geranyl pyrophosphate). Fpp and GGpp are essential for post-translational protein prenylation and for the synthesis of ubiquinone, which is implicated in the mitochondrial respiratory chain and the production of ROS (Figure 1C). In agreement with EGR2 being downstream of EWSR1-FLI1, we observed a highly significant overlap between EWSR1-FLI1- and EGR2-regulated genes (Appendix A). The pathway enrichment analysis of down-regulated genes confirmed that EWSR1-FLI1 and EGR2 regulate MVA pathway genes in EwS (Appendix A). This finding was further extended by the RNAseq analysis of the expression of MVA pathway enzymes (Figure 1D, in blue) and prenyltransferases (Figure 1D, in green) upon EWSR1-FLI1 knock-down (EWSR1-FLI1-KD) in six Ewing cell lines (A673, SKNMC, TC71, CHLA10, MHHES1, and EWS1), as well as in the A673/TR/shEF1 DOX-inducible system (shEF1) [18]. This analysis showed that 7 out 16 enzymes were consistently down-regulated upon EWSR1-FLI1-KD (Figure 1D). Using a CRISPR/Cas9 approach, an EWSR1-FLI1 translocation was recently generated in primary Mesenchymal Stem Cells (MSCs) isolated from an Ewing patient [19]. As shown in Figure 1D, MVA pathway genes were up-regulated in EWIma1 cells as compared with the parental MSCs and were down-regulated in the EWlma1 clone upon EWSR1-FLI1-KD. ChIPseq experiments further documented a direct binding of EGR2 to the promoter regions of HMGCS1 and HMGCR, two essential genes of the MVA pathway, as well as to promoters of MVK and MVD (Figure 1E, Appendix A).

These findings were confirmed at the protein level, showing that EWSR1-FLI1 and EGR2-KD led to a dramatic decrease in the expression of enzymes of the MVA pathway in four different EwS cell lines (Figure 2). Altogether, these data indicate that EWSR1-FLI1 regulates the expression of key mevalonate pathway enzymes via EGR2.

### 3.2. Ewing Sarcoma Cells Are Dependent on the Mevalonate Pathway

The observation that EWSR1-FLI1 and EGR2 regulate enzymes of the MVA pathway prompted us to investigate whether Ewing cells’ growth may be dependent upon this pathway. We interrogated the DepMap portal [26], which reports the genome-wide screening of dependencies by CRISPR/Cas9 invalidation in 1054 cell lines corresponding to 27 different cancer types, including 25 different Ewing cell lines [27,28,29,30]. As shown in Figure 3A, we noticed that 9 out of 16 genes of the MVA pathway (including the 5 prenyltransferases) demonstrated median dependency scores lower than the threshold value of −0.5, suggesting that EwS cells are strongly dependent upon the MVA pathway. Moreover, in agreement with the MVA pathway being regulated by the EWSR1-FLI1/EGR2 axis, the global expression of 16 MVA pathway enzymes, and specifically the 2 rate-limiting genes HMGCR and HMGCS1, was significantly higher in Ewing cell lines as compared with other cancer cell lines (Figure 3B and Appendix A).

In parallel to the investigation of EGR2 downstream signaling, we also performed a medium-throughput drug-repositioning screening using a set of more than 1200 off-patent drugs in a microscopy image-based assay. We used one dose of treatment (10 µM) and two time points (24 h and 72 h) to investigate the sensitivity of the A673 Ewing cell line to these compounds (Appendix A). The selection criteria of cytotoxic drugs in A673 cells are described in detail in Appendix A. In total, 187 compounds were identified as positive hits from the screening as inducing a significant decrease in Ewing cell growth at 24 and/or 72 h of treatment. Among these, 110 compounds could be assigned to 10 functional and structural families (Figure 3C, Appendix A). As expected, given the sensitivity of Ewing cells to chemotherapy, we observed an important enrichment of drugs classified as “Oncology and Antineoplastic” therapeutic drugs (22%), including the inhibitors of topoisomerases class II used in the treatment of Ewing sarcoma, such as Doxorubicin and Etoposide, as well as compounds regulating DNA synthesis via the folate cycle (Amethopterin and Methotrexate) or purine synthesis (Appendix A). The Enolid family, which includes regulators of the Na+/K+ pump, also showed up in this screening (Appendix A). Interestingly, the overexpression of the ATP1A1 subunit (ATPase Na+/K+ transporting subunit alpha 1) of the Na+/K+ pump was recently reported as a downstream target of EWSR1-FLI1 [31]. Of particular interest was the observation that five out of the six tested statins, which are competitive inhibitors of HMGCR, a rate-limiting enzyme of the MVA pathway, demonstrated cytotoxicity in the A673 Ewing cell line at 72 h (Figure 3C). The only statin that did not show up in this screening was the hydrophilic pravastatin, which is known to require an organic anion transporter (SLCO1B1) specifically expressed in hepatocytes but not in Ewing cells. Interestingly, the results from this screening are in full agreement with the DepMap database of pharmacological dependencies [32], whereby Ewing sarcoma cell lines demonstrated the highest sensitivity to Atorvastatin and Simvastatin (Figure 3D) as compared with cell lines from other malignancies. This prompted us to further define the IC50 of three statins (Atorvastatin, Simvastatin, and Lovastatin) for four Ewing cell lines and four primary cultures from Ewing PDX, as well as four non-Ewing cell lines (Figure 3E and Appendix A, Table 1). All Ewing cell lines as well as primary cultures from PDX tumors exhibited an IC50 in the micromolar range. On the contrary, apart from the malignant rhabdoid tumor cell line KD, other control cell lines demonstrated much higher IC50 values, in the 10–100 micromolar range (Table 1). Lovastatin, which did not show up in the DepMap screening, also demonstrated growth inhibitory properties in our analysis of Ewing cells. Altogether, these results indicate that two completely independent strategies, one based on the functional analysis of EGR2, a critical target of EWSR1-FLI1, and the other based on an agnostic screening of Ewing cells’ pharmacological vulnerabilities, point out the MVA pathway as a major dependency in Ewing sarcoma.

### 3.3. Atorvastatin Impacts Cell Proliferation of Ewing Cell Lines through Altered Anchorage on the Membrane of Small G-Proteins and Increased Intra-Cellular ROS

As all three tested statins give very similar results on Ewing cell lines, we selected Atorvastatin to assess its impact on cell-cycle progression and on apoptosis in three Ewing cell lines. No significant effects on the cell cycle were observed at the IC50. We therefore increased concentrations to 2× IC50, 1 µM for A673 cells, 2.6 µM for TC71, and 2.4 µM for POE. Again, only modest effects were observed: Atorvastatin treatment for 72 h had no significant effects on the different cell-cycle phases of A673 and TC71 cells and significantly reduced the proportion of S-phase cells for POE with a parallel increase in G0/G1 cells (Figure 4A).

However, in all three cell lines, we observed that Atorvastatin induced a decreased EdU incorporation, suggesting that this compound may induce a reduced incorporation of nucleotides during the S-phase (Figure 4B). To further explore this hypothesis, we performed EdU pulse–chase experiments and monitored S-phase progression at 0, 2, 4, 6, 8, and 10 h. We thus observed delayed S-phase entry and exit in Atorvastatin-treated cells (Figure 4C,D and Appendix A). To further confirm these results, we synchronized cells at the G1/S boundary with double thymidine treatment (Appendix A). After release of thymidine, we observed a delay of around 2 h in the progression into the S-phase of cells treated with Atorvastatin (Appendix A). Atorvastatin also induced a strong increase in p21 (CDKN1A) expression (Figure 4E). These data strongly suggest that the inhibition of growth by Atorvastatin induces an inhibition of cell proliferation by acting on the DNA replication velocity. We also observed that Atorvastatin induced an increase in Annexin-V-positive cells, as well as increased levels of cleaved Caspase 3 and cleaved PARP forms, as compared with DMSO-treated cells (Figure 4F). As the expected effect of statins is to inhibit HMGCR by competitive inhibition of HMG-CoA (Figure 1C), we reasoned that statin inhibition of HMGCR should be rescued by supplementing the growth medium with downstream products of the MVA pathway. We hence evaluated the ability of mevalonate acid (MEV), geranylgeranyl pyrophosphate (GGpp), or farnesyl pyrophosphate (Fpp) to rescue the Atorvastatin-induced growth inhibition. Highly significant rescues of cell growth (Figure 4G) and of p21 induction (Figure 4E) were observed in three different Ewing cell lines with the exception of A673, the cell growth of which was not rescued by Fpp. The slowdown of S-phase progression by Atorvastatin was also rescued by MEV (Figure 4H). Altogether, these data show that the inhibition of HMGCR mostly accounted for the statin-induced cell growth decrease.

The products of the MVA pathway play important roles in protein prenylation, a post-translational modification essential for the biological activities of numerous proteins. Indeed, prenyltransferases catalyze the addition of isoprenoid, farnesyl, or geranylgeranyl to the C-terminus of proteins (Figure 1C), thus creating a hydrophobic domain and enabling protein anchorage on the membrane. Small G-proteins, including the RAS, RHO, and RAC families, are dependent on prenylation for their localization to cell membranes and for their activity in cellular processes, including cell proliferation and migration (for a review, see [34,35]). To assess the impact of Atorvastatin on protein prenylation, we studied the expression of the prenylated forms of RHOA, RAS, and RAC1 by Western blot after extraction with the triton X-114 fractioning method. We observed that Atorvastatin induced a decrease in the membrane-bound forms of these proteins that can be rescued by MEV (Figure 5A).

The MVA pathway is a major cellular redox regulator that acts on the biosynthesis of ubiquinone (CoQ10) implicated in the regulation of intra-mitochondrial reactive oxygen species (ROS) via the respiratory chain (for a review, see [36]) and on the maturation of glutathione peroxidase 4 (GPX4) by modulating lipid peroxidation and the ferroptosis process (for a review, see [37]) (Figure 5B). Excessive ROS generation can result in DNA damage and programmed cell death, including apoptosis. In Ewing sarcoma, it has been demonstrated that the balance of intracellular ROS is tightly regulated [38,39,40]. To elucidate whether ROS generation may be involved in the cytotoxic effect of Atorvastatin on Ewing cell lines, we first quantified the intracellular ROS levels by flow cytometry and observed that Atorvastatin treatment induced a dose-dependent increase in intracellular ROS in three Ewing cell lines (Figure 5C). We then showed that this increase can be rescued by MEV and GGpp but not by Fpp in A673 cells (Figure 5D). As Fpp cannot rescue Atorvastatin-induced growth inhibition either (Figure 4G), this suggests a correlation between increased ROS production and cell growth inhibition. To explore this correlation, we performed rescue experiments with antioxidant N-Acetyl Cysteine (NAC) (Figure 5E). In agreement with increased ROS production being responsible for altered cell growth, NAC treatment both abolished the increased intra-cellular ROS (Figure 5E, bottom) and significantly rescued the growth inhibition induced by Atorvastatin treatment (Figure 5E, top).

To investigate whether increased ROS levels may lead to DNA-damage-induced ROS, we analyzed the expression of γH2AX in A673 cells after 72 h of Atorvastatin treatment in the absence or presence of MEV, Fpp, and GGpp. Atorvastatin led to a dramatic increased expression of γH2AX that could be rescued by MEV and GGpp but not Fpp (Figure 5F). Moreover, NAC treatment also allowed a partial reversion of γH2AX induction by Atorvastatin (Figure 5G). The MVA pathway end product Isopentyl-pp is also essential for the maturation of GPX4, a key enzyme to limit the peroxidation of lipids and ferroptosis (Figure 5B). We assessed this function with the bodipy fluorescent probe, which enables the detection of free-radical-mediated lipid peroxidation. We observed that Atorvastatin treatment induced a dose-dependent increase in lipid peroxidation in three Ewing cell lines (Figure 5H) that was rescued by MEV and GGpp but, again, not by Fpp in A673 cells (Figure 5I).

Altogether, these data show that statins induce pleiotropic effects on Ewing cells as a consequence of the inhibition of the MVA pathway, including altered protein prenylation, as well as increased intracellular ROS and lipid peroxidation.

### 3.4. Atorvastatin Induces a Reduction in Ewing Tumor Growth In Vivo

Finally, we assessed the impact of Atorvastatin on EwS growth using the 3D spheroid assay and Ewing PDX models. As shown below, using in vitro 2D assays (Figure 3E and Appendix A), a dose-dependent decreased growth of Ewing 3D spheroids was observed with cells from four Ewing PDX tumors upon Atorvastatin treatment (Figure 6A). Moreover, when treating three Ewing sarcoma PDX models (IC-pPDX-87, IC-pPDX-3, and IC-pPDX-164) with Atorvastatin at doses (10 mg/kg) calculated to correspond to those used in humans to treat hypercholesterolemia [41], we observed a significant growth reduction of these tumors as compared with those of the control group (Figure 6B).

## 4. Discussion

In this manuscript, we show that the MVA pathway is a critical vulnerability of Ewing sarcoma as shown by two independent approaches. Firstly, we demonstrate that, via the regulation of Ewing-susceptibility gene EGR2, EWSR1-FLI1 up-regulates various genes encoding enzymes of the pathway. Secondly, we show that statins, which inhibit HMGCR, a key rate-limiting enzyme of the pathway, decrease cell proliferation and induce apoptosis of Ewing cells at much lower IC50 values than in other tumor types tested here or described in the literature. Indeed, extensive analyses of these publications report that IC50 values surpass 10 micromolar in most cancers, such as breast cancer [42], glioma [43], and pancreas cancer [44]. The sensitivity of Ewing cells to statins is in full agreement with data from large CRSPR/Cas9-based and pharmacological dependency screenings [26,27,28,32].

Some large retrospective studies or meta-analyses of cancer patients have reported a significant decrease in cancer incidence or an increased survival in statin-treated patients, whereas others have failed to detect the significant impact of statins on cancer evolution (reviewed in [45,46]). These studies suggest that different factors may account for the inconsistency in the published results, including variability of follow-up periods, heterogeneity of the investigated populations in terms of types of cancer and treatments, diversity of the statin compounds, dosage, and treatment duration, as well as many inherent biases of such retrospective or meta-analyses. While a number of biomarkers has been proposed to account for statin sensitivity, so far, none of these markers have been shown to be robust enough for patient stratification. There is a general agreement that the MVA pathway is an important tumor vulnerability and that statins constitute the most promising method to block this pathway and should be seriously considered within the anti-cancer drug arsenal. To evaluate this, 58 clinical trials are currently open to more precisely evaluate statins in cancer prevention or treatment (https://clinicaltrials.gov; accessed on 6 May 2022).

The present study therefore constitutes a biologically driven indication that the EWSR1–ETS fusion, which is pathognomonic of Ewing sarcoma, constitutes a strong predictive biomarker of statin sensitivity. Interestingly, some recent reports investigated bisphosphonates, another class of MVA pathway inhibitors in the treatment of bone cancer in children. Bisphosphonates are widely used medications to treat bone resorption associated with osteoporosis or with primary or secondary bone tumors. They act by inhibiting the farnesyl diphosphate synthase (FDPS), an enzyme of the MVA pathway that is necessary for osteoclast ruffled-border formation. Interestingly, bisphosphonate and, particularly, Zoledronic acid have been shown to demonstrate anti-proliferative and anti-invasive properties in Ewing cells in vitro and in vivo [47,48,49,50]. Though it was not shown in these reports that zoledronic acid acted by inhibiting the MVA pathway, these observations converge with ours to point out the MVA pathway as an important dependency in Ewing sarcoma.

Our observation that the statin effect can be rescued by end products of the MVA pathway clearly points out that the inhibition of this pathway is the key effect of statin. It is interesting to mention that GGpp is generally more efficient than Fpp in rescuing the statin effect. This result is consistent with previous reports showing that the cell death induced by Lovastatin is rescued only by GGpp, suggesting that the genarylgeranylation of proteins is more essential than farnesylation for cell viability [51,52,53,54]. One hypothesis to account for the better rescue effect of GGpp as compared with Fpp is that isopentenyl-pp, an end product of the MVA pathway, is required for the conversion of Fpp to GGpp. Specific GGpp prenylations may thus not be achieved in the presence of Fpp alone. Reciprocally, it has been shown that GGpp can overcome the effect of farnesyl-transferase inhibitors for the prenylation of RAS [55,56]. These data therefore suggest that GGpp is more efficient in complementing Fpp deficiency than the reverse.

We further observed that the inhibition of the MVA pathway had pleiotropic effects on Ewing cells. We observed decreased cell proliferation characterized by a slowdown of the S-phase. Different mechanisms can contribute to this effect on the cell cycle. It may constitute a downstream effect of altered signal transduction. Indeed, GGpp and Fpp, two major downstream products of the MVA pathway, are critical for the isoprenylation of a variety of signal transduction molecules that require anchoring to the plasma membrane. We demonstrated such an altered anchorage for RAC1, RHOA, and RAS, but the anchorage of many other membrane receptors, including GPCRs and growth factor receptors that play a role in the transduction of proliferation signals, is also expected to be impaired. In this regard, it has been shown that RHOA is the direct geranylgeranylated effector signaling to the YAP/TAZ pathway [57], which plays important roles in the tumorigenesis of Ewing sarcoma (for a review, see [58]). The RHO GTPase–MVA pathway–YAP/TAZ axis has also been shown to be required for the proliferation and/or self-renewal of breast cancer and leukemia cells [59,60].

We also show that the cell-cycle effect is, at least in part, a consequence of increased ROS production. The MVA pathway is essential to the synthesis of coenzyme Q (CoQ10 or Ubiquinone), an essential electron carrier in the electron transport chain that contains a benzoquinone ring linked to a 10-isoprenoid-unit chain. It has recently been shown that statin treatment compromises oxidative phosphorylation and causes severe oxidative stress in mouse KPC cells and in human prostate cancer cells [61]. It has also been suggested, though not yet formally proven, that increased ROS levels may be a cause of statin-induced myopathy. We show that the statin-induced increased ROS level is a direct consequence of the altered MVA pathway, as it can be fully rescued by mevalonate acid or by GGpp. Moreover, ROS levels, increased H2AX, and proliferation defects can, at least in part, be rescued by the NAC antioxidant. Our results showing that statins induce a dramatic slowdown of the S-phase are therefore fully consistent with data demonstrating that oxidative stress alters the dynamics of DNA replication forks during the S-phase [62].

Decreased proliferation is not the sole effect of statin treatment on Ewing cells. Indeed, we observed a significant increase in apoptosis, assessed by increased Annexin V staining, cleaved Caspase 3, and PARP, upon treatment of Ewing cells with statins, in agreement with the many studies that described statins specifically triggering the apoptosis of tumors cells via the inhibition of the MVA pathway (for a review, see [45]).

Finally, and very importantly, our data also demonstrate that, as a single agent, statin treatment significantly inhibited tumor growth in three different PDX Ewing models at doses similar to those used in humans to treat hypercholesterolemia.

Altogether, our data show that the activation of the mevalonate pathway in Ewing sarcoma is a direct consequence of the activation of EGR2 by EWSR1-FLI1 and that this pathway can be efficiently targeted by statins. These FDA-approved, well-tolerated, and inexpensive compounds can hence be added to the therapeutic arsenal used in the treatment of EwS, a deadly disease for which new therapeutic approaches are urgently needed. Phase I dose-escalation studies have shown that statins are well tolerated at doses much higher than typically prescribed for cholesterol management, at least for defined periods of time, suggesting that dosage optimization could be searched for in the treatment of EwS. Even more importantly, new combination therapies can now be considered. The observation that statins induce ROS and DNA damage, as shown by increased H2AX, together with recent reports pointing out a high level of replication stress in Ewing cells [63], may stimulate research on combinations with other DNA-damaging agents or DNA-repair-inhibiting drugs [64]. In this respect, it is interesting to mention that EWSR1-FLI1 was found as a strong predictive biomarker of in vitro sensitivity to PARP inhibitors, suggesting that the combo of statins with these inhibitors may constitute an attractive therapeutic strategy.

## 5. Conclusions

We show that the mevalonate pathway is essential in Ewing sarcoma and could be a potential new therapeutic target. In fact, the EWSR1-FLI1 oncogenic protein regulates the expression level of MVA pathway proteins by its direct effect on the EGR2 protein. The statins, which inhibit HMGCR, a key rate-limiting enzyme of the MVA pathway, decrease cell proliferation and induce the apoptosis of Ewing cells in vitro and the tumor growth of PDX Ewing models.

## Figures and Tables

**Figure 1 cancers-14-02327-f001:**
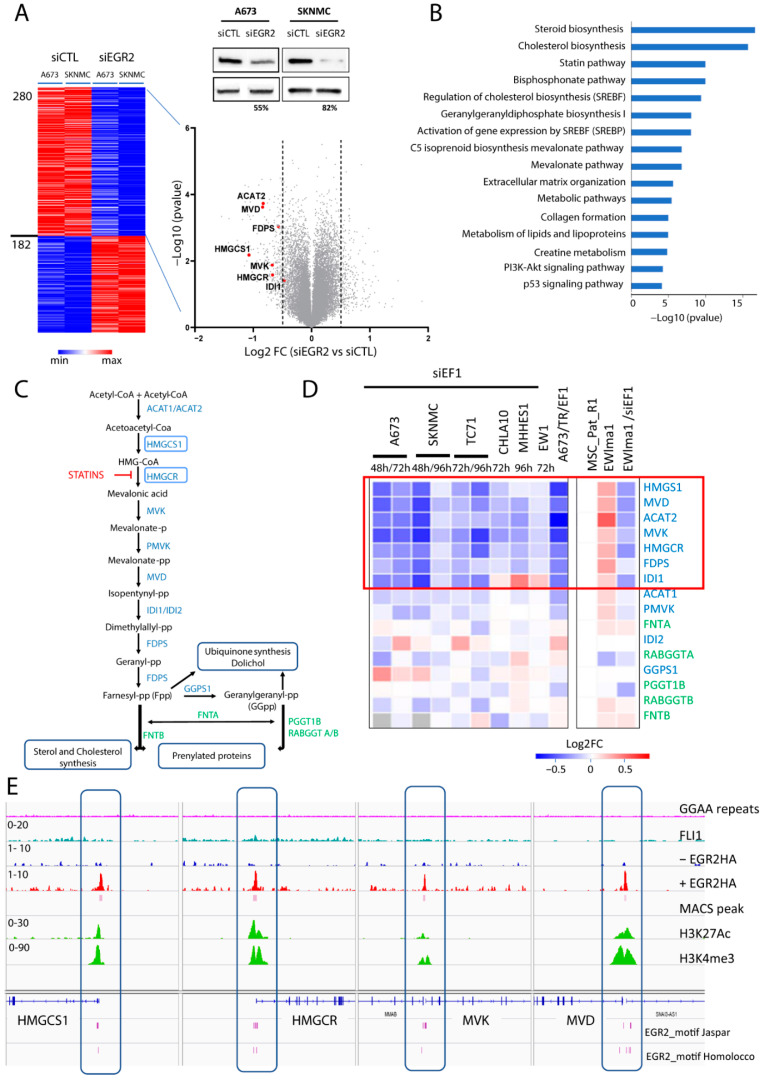
EGR2 regulates the expression of mevalonate pathway enzymes in Ewing cells. (**A**) Heat map of differentially expressed genes in the A673 and SKNMC Ewing cell lines obtained from the mean of two siEGR2 (#1 and #2) (absolute log2FC > 0.5) and Western blot analysis of EGR2 expression cells for 48 h after siEGR2 or siCTL transfection. The volcano plot shows the differentially expressed genes in A673 and SKNMC Ewing cells upon siEGR2 vs. siCTL transfection. The dotted line corresponds to the absolute value of log2FC = 0.5 and the red dots point out mevalonate pathway genes down-regulated by EGR2-KD. (**B**) Gene Ontology and pathway analysis using the Topp-Fun Gene portal performed on the 280 genes down-regulated upon EGR2-KD with a log2FC < −0.5 comparing siEGR2 vs. siCTL. (**C**) Schematic representation of the mevalonate pathway. The blue boxes indicate the 2 rate-limiting enzymes. (**D**) Heat map representation of the differential expression of mevalonate pathway enzymes (in blue) and prenyltransferases (in green) in 6 Ewing cell lines after invalidation with siRNA targeting EWSR1-FLI1 (siEF1) for 48 h (A673 and SKNMC), 72 h (A673, TC71, CHLA10, and EW1), and 96 h (SKNMC, TC71, and MHHES1) and in the A673/TR/shEF1 inducible system (shEF1) treated with Dox for 7 days [24] and parental MSC cells (MSC_Pat_R1) as well as in EWIma1 cells invalidated or not for 72 h by siEF1. EWIma1 was generated by engineering an EWSR1-FLI1 translocation in a Mesenchymal Stem Cell (MSC_Pat_R1) [19]. The red box shows the 7 enzymes commonly up-regulated by EGR2 and EWSR1-FLI1. (**E**) CHIP-seq profiles for EWSR1-FLI1, EGR2, H3K27Ac, and H3K4me3 binding to HMGCS1, HMGCR, MVK, and MVD promotors in the POE shEGR2_4#22 clone [13] expressing a tagged EGR2 (EGR2HA). The endogenous EGR2 was invalidated by shRNA induced upon Dox treatment (+EGR2HA) or not (−EGR2HA).

**Figure 2 cancers-14-02327-f002:**
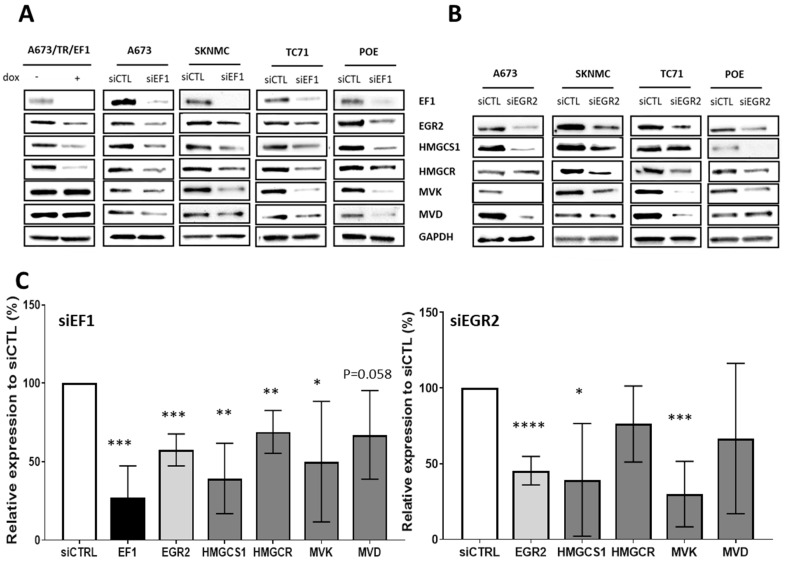
EWSR1-FLI1 and EGR2 regulate the protein expression of the mevalonate pathway in Ewing cells. (**A**,**B**) Western blot analysis of HMGCS1, HMGCR, MVK, and MVD expression in total lysates of the A673, SKNMC, TC71, and POE Ewing cell lines transfected with a control siRNA (siCTRL) or with siRNA targeting EWSR1-FLI1 (siEF1) (**A**) or EGR2 (siEGR2#1) (**B**) for 72 h. The A673/TR/shEF1 inducible cellular model was treated with doxycycline for 7 days (+Dox) to invalidate the EWSR1-FLI1 expression or was untreated (−Dox). (**C**) Quantification of protein expression levels of EWSR1-FlI1, EGR2, HMGS1, HMGCR, MVK, and MVD after invalidation by siRNA of EWSR1-FLI1 (siEF1) or EGR2 (siEGR2#1) for 72 h; the graph represents the relative expression to siCTRL for each cell line. The histograms correspond to the mean value of all Ewing cell lines (n = 4). Data are presented as mean +/− SD. **** *p* value < 0.0001, *** *p* value < 0.001, ** *p* value < 0.01, * *p* value < 0.05 versus siCTRL.

**Figure 3 cancers-14-02327-f003:**
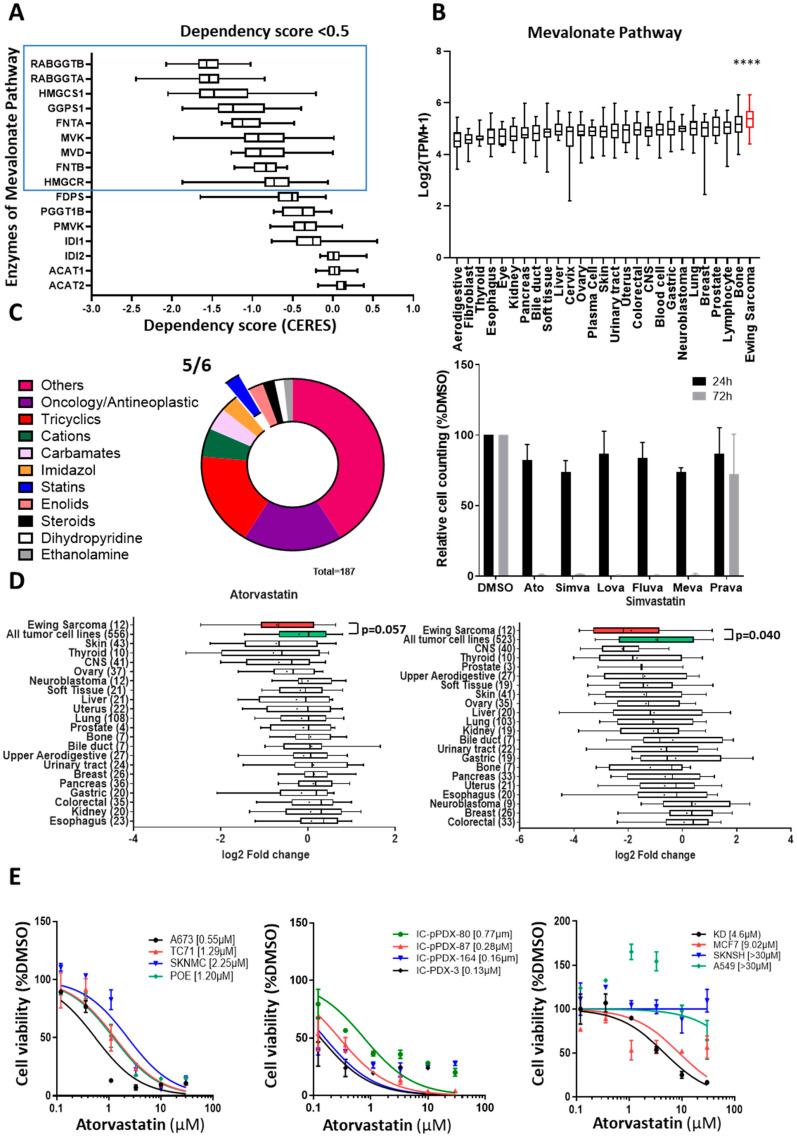
Ewing sarcoma tumors and cell lines are sensitive to statin treatment. (**A**) Boxplot of median dependency scores for mevalonate pathway enzymes and prenyltransferases in 25 Ewing sarcoma cell lines upon CRISPR lethality screening (DepMap portal, Avana library). The blue box highlights genes with a median dependency score < −0.5 in Ewing cell lines. (**B**) Comparison of the mean expression level of mevalonate pathway enzymes and prenyltransferases across various tumor cell lines (DepMap portal) [33]. Data are presented as min-to-max boxplots with median; **** *p* value < 0.0001, unpaired Student’s *t*-test with Welch’s correction. (**C**) Results of the high-throughput compound screening using the Prestwick library with one dose of treatment (10 µM) and two time points (24 h and 72 h) using the A673 Ewing cell line. Among the 187 hit compounds, 110 compounds can be grouped into 10 different functional and structural families. A total of 5 out of 6 statins available in the Prestwick library induced a decrease in cell proliferation after 24 h of treatment and demonstrated major cytotoxicity at 72 h. (**D**) Boxplots of the log2FC of fluorescent intensity of the different tumor groups upon Atorvastatin and Simvastatin treatment normalized to DMSO treatment. The red boxplots correspond to Ewing cell lines and the green boxplots to the mean of all other tumors. Data are represented as 10–90% percentiles with median (line) and mean (dot) obtained with Bayesian Student’s *t*-tests (data from the DepMap portal). (**E**) Dose–response curves and IC50 values of Atorvastatin for 4 Ewing cell lines (A673, TC71, SKNMC, and POE), 4 dissociated cells from Ewing PDX (IC-pPDX-80, IC-pPDX-87, IC-pPDX-164, and IC-pPDX-3), and 4 non-Ewing cell lines (KD, malignant rhabdoid tumor; SKNSH, neuroblastoma; MCF7, breast cancer; A549, lung carcinoma). Cell viability was quantified by cell counting with trypan blue exclusion after 72 h of treatment with 6 doses of atorvastatin and expressed as the percentage relative to cells treated with DMSO (n = 2).

**Figure 4 cancers-14-02327-f004:**
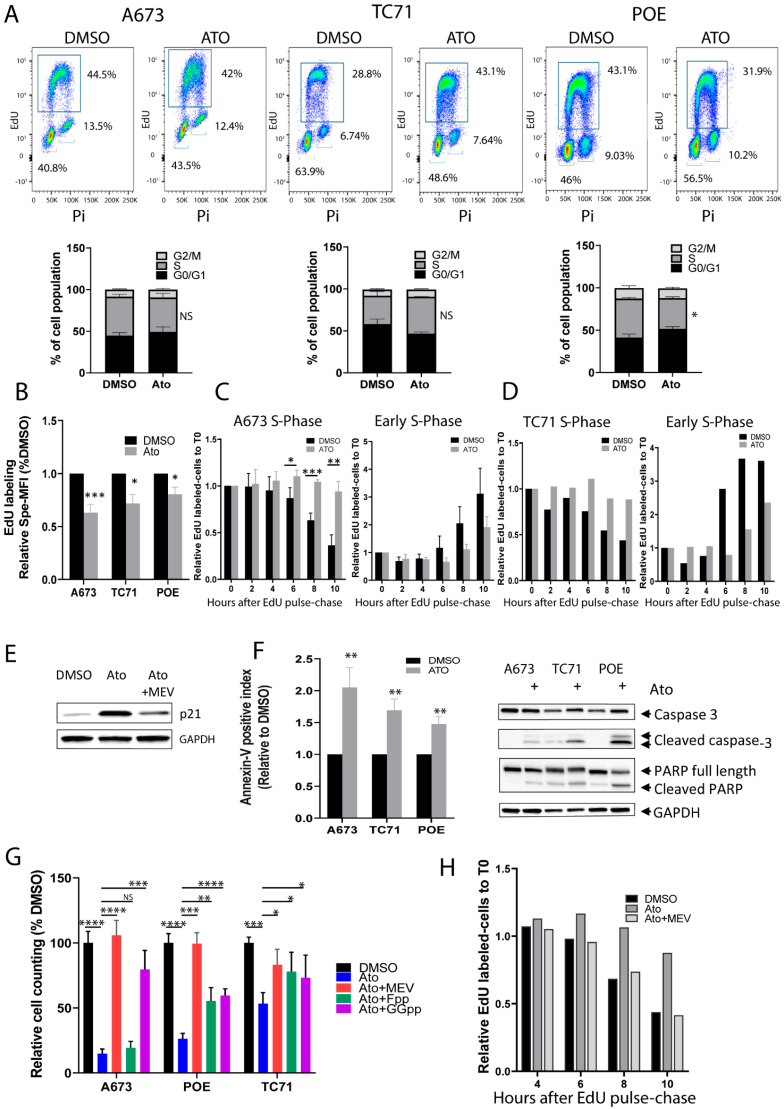
Atorvastatin induces an MVA pathway-dependent slowdown of the S-phase of cell cycle and apoptosis. (**A**) Analysis of cell cycle by EdU staining in A673 (n = 3 biological replicates), TC71 (n = 2 biological replicates), and POE (n = 3 biological replicates) Ewing cells with X2 IC50 doses of Atorvastatin for 72 h. (**B**) Atorvastatin induces a decrease in EdU incorporation during the S-phase. The graph represents specific MFI of EdU-labeled cells relative to DMSO for each cell line (n > 3 biological replicates). (**C**,**D**) Analysis of cell-cycle progression by EdU pulse–chase method in the A673 (n = 3 biological replicates) (**C**) and TC71 (n = 1) (**D**) cell lines treated, respectively, with 1 µM and 5 µM Ato for 72 h (80% of proliferation inhibition). Cells were collected 2, 4, 6, 8 h, and 10 h after EdU pulse. The histograms represent the EdU-labeled cells relative to T0 (start point after EdU pulse) during the S-phase and early S-phase. (**E**) Analysis of p21 expression in the A673 cell line treated with 0.5 µM Atorvastatin for 72 h and with rescue by MEV (100 µM). (**F**) Study of apoptosis by Annexin-V-staining assay and by assessing the presence of cleaved caspase-3 and cleaved-PARP by Western blot in 3 Ewing cell lines (A673, TC71, and POE) treated for 72 h with 0.5 µM (IC50), 2.6 µM (IC50 × 2) and 2.4 µM (IC50 × 2) Atorvastatin, respectively (n > 5 biological replicates). (**G**) Measurement of cell viability 72 h after Atorvastatin treatment at 2 × IC50 (A673, 1 µM; TC71, 2.6 µM; POE, 2.4 µM) with rescue by Mevalonic acid (MEV) (100 µM), Farnesyl pyrophosphate (Fpp) (10 µM), or GeranylGeranyl-pyrophosphate (GGpp) (10 µM) metabolites (n = 3 biological replicates). Data are presented as mean +/− SEM (A,B,F) or +/− SD (C,G). **** *p* value < 0.0001, *** *p* value < 0.001, ** *p* value < 0.01, * *p* value < 0.05 versus DMSO (A,B,F,G) or versus Ato treatment (**G**), unpaired Student’s *t*-test. (**H**) Atorvastatin induces a decreased EdU incorporation during the S-phase with rescue by MEV (100 µM) in the A673 cell line. The barplot represents the EdU-labeled cells relative to T0 (start point after EdU pulse) during the S-phase 4, 6, 8, and 10 h post EdU pulse (n = 1).

**Figure 5 cancers-14-02327-f005:**
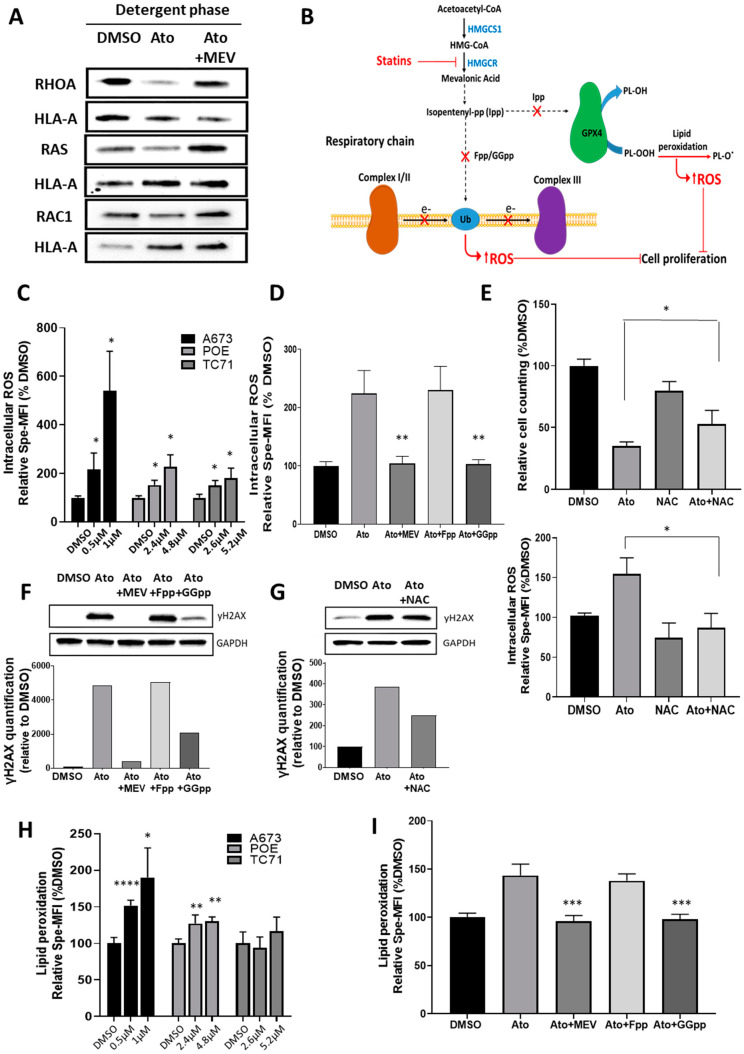
Altered protein prenylation, lipid peroxidation and increased intracellular ROS upon statin treatment. (**A**) Western analysis of tree main prenylated proteins (RHOA, RAS, and RAC1) of the mevalonate pathway after treatment with Atorvastatin (0.5 µM) and with rescue by Mevalonic acid (MEV) (100 µM) in the A673 cell line. MHC1 protein membrane (HLA-A) was used as control of the partitioning of membrane proteins. (**B**) Schematic representation of statins impact on mevalonate pathway, intracellular ROS and lipid peroxidation. (**C**–**E**) Atorvastatin induced increased intracellular ROS in a dose-dependent manner through mevalonate pathway. (**C**) The graph represents specific MFI using CellRox probe in the A673, POE, and TC71 Ewing cell lines treated with 2 doses of Atorvastatin (A673 0.5 µM and 1 µM, POE 2.4 µM and 4.8 µM, TC71 2.6 µM and 5.2 µM) (n = 3 biological replicates). (**D**) The A673 Ewing cell line was treated for 72 h with Atorvastatin at the IC50 (0.5 µM) alone or with mevalonic acid (MEV) (100 µM), Farnesyl pyrophosphate (Fpp) (10 µM), or GeranylGeranyl-pyrophosphate (GGpp) (10 µM) metabolites (n = 3 biological replicates). (**E**) Measurement of cell viability (**Top**) and intracellular ROS (**Bottom**) after treatment with Atorvastatin (IC50 × 2 = 1 µM) for 48 h associated with NAC (3 mM), an inhibitor of intracellular ROS in the A673 cell line (n = 3 biological replicates). (**F**,**G**) Analysis and quantification of γH2Ax expression by Western blot of the A673 cell line treated with 0.5 µM Atorvastatin for 72 h and rescued by MEV (100 µM), Fpp (10 µM), or GGpp (10 µM) (**F**), or by treatment with NAC (3 mM) (**G**). (**H**) The graph represents specific MFI using a Bodipy probe in the A673, POE, and TC71 Ewing cell lines treated with 2 doses of Atorvastatin (n = 3 biological replicates). (**I**) The A673 cell line was treated with 0.5 µM Atorvastatin alone or with MEV (100 µM), Fpp (10 µM), or GGpp (10 µM) (n = 3 biological replicates). Data are presented as mean +/− SD. **** *p* value < 0.0001, *** *p* value < 0.001, ** *p* value < 0.01, * *p* value < 0.05 versus DMSO (**C**,**H**) or versus Ato treatment (**D**,**E**–**I**), unpaired Student’s *t*-test.

**Figure 6 cancers-14-02327-f006:**
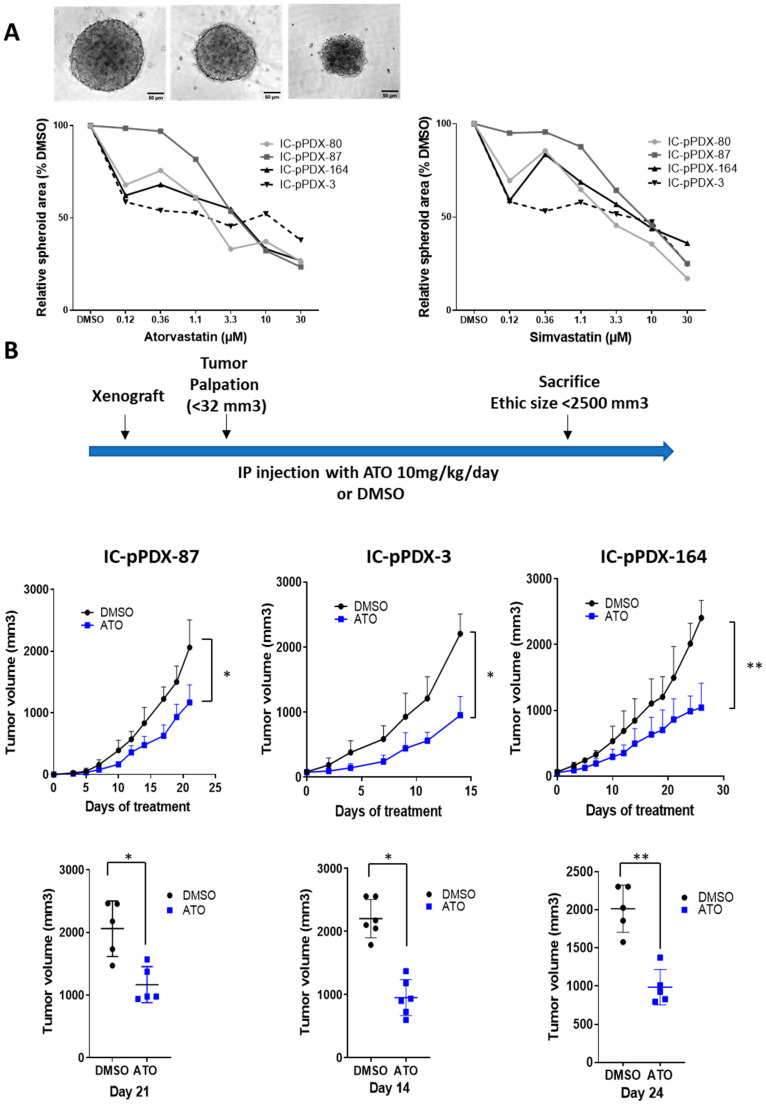
Statins induce a decrease in 3D spheroid growth and impair tumor growth of Ewing sarcoma PDX. (**A**) Representative images and dose–response curves showing the spheroid area relative to DMSO measured with image J software of 4 dissociated Ewing PDX tumors (IC-pPDX-80, IC-pPDX-87, IC-pPDX-164, and IC-pPDX-3) treated by increasing the dose of Atorvastatin. Scale bar represents 50 μm. (**B**) Experimental design of IP injection of Atorvastatin (10 mg/kg/day) or DMSO in Ewing PDX and tumor growth curves and tumor volumes at ethical sacrifice (day 21 for IC-pPDX-87, day 14 for IC-pPDX-3, and day 26 for IC-pPDX-164). Values represent the mean of tumor volume +/− SD (n = 5–6 mice per group). ** *p* value < 0.01, * *p* value < 0.05, determined with Mann–Whitney test for tumor volume and paired Student’s *t*-test for growth curve.

**Table 1 cancers-14-02327-t001:** IC50 for Atorvastatin, Simvastatin, and Lovastatin in the different Ewing cell lines. (RT: Rhabdoid Tumor, BC: Breast Cancer, NB: neuroblastoma).

	Ewing Cell Lines				Short Term Culture of Ewing PDX				Other Cell Lines			
IC50 (µM)	A673	TC71	SKNMC	POE	IC-pPDX-80	IC-pPDX-87	IC-pPDX-164	IC-pPDX-3	KD	MCF7	SKNSH	A549
									RT	BC	NB	Lung
Atorvatatin	0.55	1.29	2.25	1.2	0.75	0.3	0.2	0.13	4.6	9.02	no effect	124.6
Simvastatin	0.34	1.03	2.6	0.2	1.08	0.39	0.24	0.11	1.51	12.62	62.55	17.54
Lovastatin	1.06	3.26		3.98	0.69	0.83	1.18	0.55	5.03	7.39	43.13	26.05

## Data Availability

All data associated with this study are present in the paper or in the Appendix A.

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
