# Peer review of "Upregulation of the Mevalonate Pathway through EWSR1-FLI1/EGR2 Regulatory Axis Confers Ewing Cells Exquisite Sensitivity to Statins"

_cancers, 2022, doi:10.3390/cancers14092327_

Round 1
Reviewer 1 Report
In the article entitled “Upregulation of the Mevalonate Pathway through an EWSR1-2 FLI1/EGR2 Regulatory Axis Confers Ewing Cells Exquisite Sensitivity to Statins” the authors propose the mevalonate pathway as a new target for the treatment of Ewing Sarcoma and analyse the effects induced by statins in in vitro and in vivo models of Ewing Sarcoma. The study is interesting, and the results might be relevant for setting novel target therapies against this sarcoma, however, the manuscript shows over-interpretation of some data, and the representation of some results is confused. Specifically, the following points should be addressed:
Major points:
1) I am not clear on how many times all the experiments were performed. Please add the missing information in the “Materials and Methods” section and/or in the figure captions.
2) Densitometric analysis shown in Figure 2C represents the mean values of all the cell lines analysed by western blotting or a single cell line used as representative?
3) Why have you decided to use the concentrations of 2xIC50 in almost the experiments? Usually, the IC50 concentration and a lower dose are used to assess if the same cellular/molecular effects are obtained with less off target effects. Moreover, you used IC50 and 2xIC50 concentration to study the same cellular process (for example cell cycle analysis by cytofluorimeter: 1 μM; p21 expression levels by western blot: 0.5 μM…) and this is not fine
4) The conclusions on cell cycle perturbations are not totally convincing:
-Cell cycle analysis in A673 cell line does not appear to show statistically significant changes (Figure 4A), indeed there are no differences of cell percentage in G1/S/G2 phases between DMSO and ATO treatment. The text should reflect this result.
-Excluding A673 cells for the above reasons, TC71 and POE cell lines show an opposite trend. How do you explain this difference?
-In Figure 4C-4D standard deviation and statistical analysis are missing.
-You should analyse p21 expression levels also in TC71 and POE cells to confirm this result. Please use the same drug concentration in FACS analysis and western blotting assay.
-In which cell line did you performed experiments shown in Figure 4H? Moreover, standard deviation and statistical analysis are missing.
-Considering all these points the conclusions on cell cycle inhibition are not well supported by the results, you should add/repeat some experiments to corroborate what you stated
Minor points:
1) The axis titles of many graphs in figures 4, 5 and 6 are too small, please increase the font size to improve the understanding of the results
2) Check the spaces between words (72h, 1μM…)
Reviewer 2 Report
In the paper “Upregulation of the mevalonate pathway through an EWSR1-FLI1/EGR2 regulatory axis confers Ewing cells exquisite sensitivity to statins”, the authors demonstrate that Ewing sarcoma cells have a marked susceptibility to statins administration due to enhanced expression levels of mevalonate pathway genes. Treatment with statins affected Ewing sarcoma cells viability, through increase of ROS levels and lipid peroxidation, cell cycle inhibition and apoptosis induction. Furthermore, they showed an anticancer effect of statins in vivo by using Ewing PDX models.
The paper by Buchou et al. is well written, the results are clearly described and the work appears well structured. The discovery of novel potential therapeutic targets is of vital importance for childhood diseases. In this line, the presented work identified statins, well-tolerated cholesterol-lowering drugs, as new attractive therapeutic strategies in combination therapies against Ewing sarcoma.
A few minor points should be addressed:
- In Fig.1D, the names of the different cell lines appear not clear. Furthermore, different time points have been analyzed across the cell lines. The authors should explain this decision, unless the time points should be standardized.
- In Fig. 2C, does the control correspond to 100%? It should be stated in the figure legend or in the graph.
- The statistical analysis is missing in the followings: Fig. 2C, 3E, 4C, 4D, 4H, 5F, 5G, 6A.
- The dimension of some images should be increased (e.g. Fig. 3D, 6A), at least the characters of text.
- Are the results in Fig. 4A significative? The percentage of cells in the different cell cycle phases appears very similar from the bars.
